



# A revised ocean mixed layer model for better simulating the diurnal variation of ocean skin temperature

Eui-Jong Kang[1], Byung-Ju Sohn[1,2], Sang-Woo Kim[1], Wonho Kim[3], Young-Cheol Kwon[4], Seung-Bum Kim[4], Hyoung-Wook Chun[4], and Chao Liu[2]

[1] School of Earth and Environmental Sciences, Seoul National University (SNU), Seoul, South Korea.
[2] School of Atmospheric Physics, Nanjing University of Information Science and Technology (NUIST), Nanjing, China
[3] Korea Institute of Atmospheric Prediction Systems (KIAPS), Seoul, South Korea
[4] Numerical Modeling Center, Korea Meteorological Administration (KMA), Daejeon, South Korea

*Correspondence to*: Byung-Ju Sohn (sohn@snu.ac.kr)

**Abstract.** Sea surface temperature (SST) is a pivotal parameter in climate, weather, and ocean sciences because it plays a decisive role in ocean-atmosphere interactions. Identifying deficiencies in the ERA5 reanalysis of ocean skin temperature, we revisited the ocean mixed layer (OML) model used at ECMWF and identified errors in the model, and revised it accordingly. Validation of the revised model was conducted through comparisons of simulated temperatures at the skin layer and 1-m depth with observations from ship-board infrared interferometers and buoy-mounted thermistors. These
comparisons revealed a strong correlation, with an absolute mean deviation of less than 0.1 K and a standard deviation under 0.5 K. We concluded that the temperature at the ocean-atmosphere interface possesses the same degree of accuracy. Moreover, the results closely align with anticipated distributions of solar radiation. Consequently, the revised OML model shows promising potential for improving the simulation of diurnal interface SST variations in weather and climate models.

## 1 Introduction

Sea surface temperature (SST), recognized as an essential climate variable of the global climate observing system (Bojinski et al., 2014), holds immense significance in the areas of climate, weather, and ocean sciences. Its influence can be realized through crucial atmospheric processes like evaporation, cloud formation, precipitation, and circulation patterns. SST exerts a substantial impact on atmospheric dynamics, playing a pivotal role in both short-term weather events and long-term climate phenomena (Manning and Solomon, 2007). Its involvement in phenomena like El Niño and La Niña drives global shifts in
weather patterns, underscoring its essential role in the earth-atmosphere climate system (Lau, 1997; McPhaden et al., 2006). Furthermore, SST's response to elevated $CO_2$ levels raises pivotal questions about its contribution to changes in the atmospheric circulation and the global hydrological cycle (Xie et al., 2010; Burls and Fedorov, 2018; Sohn et al., 2019; Zhang et al., 2023). The implications certainly extend to ocean studies, encompassing ocean-atmosphere interactions and associated chemical and biological processes (Chavez et al., 2002; Holmgren et al., 2006).



Existing methods for constructing SST data across the global oceans, currently available on a daily time scale, incorporate satellite-borne radiometric measurements and buoy/shipborne measurements (Reynolds and Smith, 1994; Ishii et al., 2005; Reynolds et al., 2007; Donlon et al., 2012; Titchner and Rayner., 2014). Nonetheless, it is important to note that these measurements pertain to specific ocean water depths – ranging from centimeters to meters for buoys and shipborne thermistors and from microns to sub-millimeters for satellite-borne infrared and microwave sensors (Donlon et al., 2002).

While these measurements provide indispensable inputs for weather and climate models and diagnostic studies, they represent temperatures at certain depths [e.g., foundation level of Donlon et al. (2012)] rather than the ocean-atmosphere interface temperature, which governs the thermodynamic coupling of the ocean and atmosphere (Webster and Lukas, 1992). This interface temperature, commonly referred to as skin temperature, is the variable essential for calculating the air-sea flux. However, because directly measuring this interface temperature presents challenges, theoretical and empirical approaches are

employed to estimate it based on measured SSTs at various depths (Fairall et al., 1996a).

These ocean skin temperatures vary diurnally in response to the solar cycle (Price et al., 1986). The diurnal variations in skin temperature, often fluctuating within a range smaller than 1 K (Fairall et al., 1996b; Ward, 2006; Wells et al., 2009) but occasionally exceeding a few degrees under conditions of weak wind and strong insolation (Donlon et al., 1999; Merchant et al., 2008), also play an important role in the heat budget. A mere 1 K error in skin temperature can lead to a substantial 27 W

$m^{-2}$ error in surface net heat flux, as evidenced in the tropical western Pacific (Webster et al., 1996). Thus, capturing this diurnal variability is essential, as skin temperature governs moisture, heat, and radiation flux that dictate ocean-atmosphere interactions (Kawai and Wada, 2007). Consequently, the accurate modeling of diurnal variation in skin temperature holds the potential to resolve the heat budgets and related phenomena.

Despite the importance of ocean skin temperature data as a potential apparatus, it is rare to find how accurate the skin

temperatures in reanalysis products or numerical weather prediction model simulations are. Some studies have evaluated the performance of ocean skin temperature in ERA5 reanalysis by comparing against SST observations from field campaigns and meteorological stations (e.g., Luo et al., 2020). However, as the representing depths of ERA5 skin temperature and other observations are different, the direct comparison and deduced results cannot be overly credited. Here, we raise questions in the diurnal variations of skin temperature in the ERA5 reanalysis – details are found in the following section 2. The ECMWF

model employs the ocean prognostic scheme (Zeng and Beljaars, 2005; ECMWF, 2016), with use of SST data as inputs [i.e., operational sea surface temperature and ice analysis (OSTIA) level 4 data; Dolon et al., 2012] for predicting the ocean skin temperature. The deficiencies found in ERA5 reanalysis raise questions regarding the chosen prognostic scheme.

Recognizing potential problems in the used scheme for generating the ocean skin temperatures, we attempt to scrutinize the root cause of anomalous features found in ERA5 (as exhibited in the following section 2), and revise the Zeng and Beljaars

(2005) scheme for the better simulations of the diurnal variation of ocean skin temperature. Results and lessons obtained in this study will eventually lead to better weather forecasting and climate simulation.



## 2 Anomalous features of ocean skin temperature found in ERA5 reanalysis

*a. Various definitions of SST*

Prior to moving into topics related to the ocean skin temperature simulation, we clarify different SSTs encountered in our
analysis. The SST carries various definitions, depending on the used methodologies and assumptions (Donlon et al., 2002).
These can often lead to unintended misunderstanding when interpreting the SST-related findings. One common definition of
SST is the temperature measured over several meters below the air-sea interface, which is referred to as bulk SST or $SST_{depth}$,
representing SST at the certain depth. For example, if a buoy-equipped thermistor measures the temperature at a depth of 1
meter, it is designated as $SST_{1m}$. Another common definition of SST is related to the temperature measured by satellite,
airplane, and ship-board infrared (IR) or microwave (MW) radiometric instruments. The IR- and MW-retrieved temperatures
are referred to as $SST_{skin}$ and $SST_{subskin}$, which represent the temperature at a depth of a few micrometers and millimeters
below the air-sea interface, respectively. This is because the depth to which radiation can penetrate the ocean is contingent
upon the instrument wavelength. There is the temperature at the air-sea interface, referred to as $SST_{int}$, but it cannot be
directly measured with current technologies. Therefore, it is common to employ the models for calculating $SST_{int}$ with given
atmospheric and oceanic environments near the ocean surface.

In numerical weather prediction (NWP) models, the variable " skin temperature" over ocean is referred to as $SST_{int}$, while
the variable "SST" indicates the foundation SST ($SST_{fnd}$), which is the temperature at a depth of approximately 5–10 m with
no diurnal variation. The NWP $SST_{fnd}$ is commonly initialized with daily composite data (Fujiwara et al., 2017), which is a
blend of satellite retrievals and conventional measurements. For estimating the diurnally varying $SST_{int}$, the NWP models
mainly include the simplified bulk-type ocean prognostic scheme as part of their numerical system. In this study, we deal
with different SSTs, and thus, it is important to be aware of differences between $SST_{int}$, $SST_{skin}$, $SST_{subskin}$, $SST_{depth}$, and
$SST_{fnd}$ of the oceanography, as well as between "skin temperature" and "SST" of the NWP model when results are
interpreted. In addition, "T" representing the temperature is used interchangeably with "SST"; for example, $SST_{int}$ and
$SST_{fnd}$ can be abbreviated with $T_{int}$ and $T_{fnd}$.

*b. Anomalous features noted in ERA5 reanalysis of ocean skin temperature*

In order to examine how ERA5 $T_{int}$ behaves from the perspective of diurnal variation, we examine the hourly variation of
ERA5 $T_{int}$, which is defined as the variation of the difference between $T_{int}$ and $T_{fnd}$. Fig. 1 shows the geographical
distribution of $T_{int}$ hourly variation (i.e., $T_{int} - T_{fnd}$), given in a 3-hour interval, over a period of one and a half days (from 12
UTC 1 January 2020 to 21 UTC 2 January 2020). This figure provides a visual representation of how the hourly $T_{int}$ varies in
one and a half days over the global ocean. It reveals significant spatiotemporal fluctuation of $T_{int}$ between two specific time
zones (10–21 UTC and 22–09 UTC). Here, such fluctuations are depicted with Fig. 1a–1d for 12–21 UTC, Fig. 1e–1h for



00–09 UTC, and Fig. 1i−1l for 12–21 UTC in following day. Specifically, the hourly variation in Fig. 1e−1h exhibits spatially much variating patterns, contrasting to much smoother features at earlier and later 12-hour periods across the global ocean.

To present more clear behavior changes in the $T_{int}$ variation at specific times, the hourly variations are compared between 09 and 10 UTC and between 21 and 22 UTC on January 1, 2020, which are found to be the time zones showing a spatial "disruption" of the hourly variation pattern (Fig. 2). At 09 UTC, patterns appear scattered on a much smaller scale. But, after one hour, at 10 UTC, the patterns show significantly smooth and large-scale features, compared to the previous time. For example, over a one-hour period in the 30°S–60°S latitudinal band, scattered patterns over the Southern Indian Ocean,
Central South Pacific, and East of New Zealand became much smoother, and localized high-variating magnitude areas disappeared. Furthermore, after 12 hours, between 21 UTC and 22 UTC, such changes appear reversed. Much-scattered patterns and localized high-variating magnitudes are returned at 22 UTC, whose spatial patterns are similar to the ones shown at 09 UTC.

Considering that hourly variations over the three-hour period within a similar spatial regime are substantial – compare 00
UTC vs. 03 UTC and 18 UTC vs. 21 UTC in Fig. 1, these large changes within an hour should not be normal (as shown in Fig. 2). In addition, such "abrupt" changes occurring at 10 UTC and 22 UTC with a 12-hour interval are far from reasonable. These abrupt changes with a 12-hour interval are found to be persistent in ERA5 since a one-and-a-half-day period randomly chosen each month over 20 years from 2001 to 2020 shows the same abnormal behaviors (not shown). Furthermore, because these hourly variations compose the diurnal variation, which should be highly correlated with solar radiation, the abrupt
changes made at 09 UTC and 21 UTC should not be rational. Therefore, these findings raise questions about the underlying procedures to produce $T_{int}$ in ERA5.

## 3 Revised prognostic scheme for $T_{int}$

Since ECMWF model employs the ocean prognostic scheme, described in Zeng and Beljaars (2005) and ECMWF (2016), for predicting the $T_{int}$, we present a revised scheme rooted in the same physical background as in those references. The
scheme has been called as an ocean mixed layer (OML) model which predicts temperature profiles by calculating the vertical heat transport (Noh et al., 2011). Thus, we denote the aforementioned prognostic scheme as the OML model.

The OML model was designed to simulate the complex temperature profile and diurnal variation resulting from energy exchange and turbulent mixing processes occurring in the upper ocean layer (approximately 0–10 m). The model comprises one level (i.e., air-sea interface) and two layers (i.e., cool skin layer, and warm layer), as in Zeng and Beljaars (2005). The
temperature at the air-sea interface is known to be approximately 0.1–0.3 K cooler than the temperature a few millimeters



below. This phenomenon is known as the cool skin effect, and the thin layer influenced by this effect is referred to as the cool skin layer (Ewing and McAlister, 1960; Saunders, 1967; Grassl, 1976). Beneath the cool skin layer, there is a layer that experiences diurnal variation and is approximately 5–10 m thick. This layer is called the warm layer. The temperature at the bottom of the warm layer ($T_{warm}$) is fixed as $T_{fnd}$ (e.g., ERA5 SST). Regarding the ERA5 OML model, Saunders (1967) and

Fairall et al. (1996b) made contributions to establishing this theoretical background, and Fairall et al. (1996b), Beljaars (1997), and Zeng and Beljaars (2005) conducted relevant numerical studies.

However, during the later stages of the OML model development, it is now recognized that a typographical error was involved in describing $T_{int}$ and appeared passed on to other studies without being corrected. In Eq. (4) of Zeng and Beljaars (2005), the temperature difference between the air-sea interface ($T_{int}$) and the bottom of the cool skin layer ($T_{cool}$) was

expressed as follows:

$$T_{int} - T_{cool} = \frac{\delta}{k_w \rho_w c_w} \left( H + E + LW + f_s\, SW \right) \tag{1}$$

In Eq. (1), δ (in meters) denotes the depth of the cool skin layer; $\rho_w$, $c_w$, and $k_w$ are the density (kg m$^{-3}$), specific heat capacity (J kg$^{-1}$ K$^{-1}$), and conductivity (W m$^{-1}$ K$^{-1}$) of ocean water, respectively; H, E, LW, and SW (all in W m$^{-2}$) are the sensible heat flux, latent heat flux, surface net longwave radiation flux, and surface net shortwave radiation flux, respectively; $f_s$

(dimensionless) represents the shortwave absorptivity at the air-sea interface. The energy source terms on the right-hand side, except for the shortwave flux (SW), mainly contribute to cooling the surface. We denote terms as Q (i.e., Q = H + E + LW). The convention for vertical fluxes here is positive downwards and negative upwards, i.e., Q is given as a negative value, whereas SW is given as a positive value.

In Zeng and Beljaars (2005), $k_w$ was defined as the thermal conductivity, although $k_w$ should be the thermal diffusivity ($h_w$ in

m$^2$ s$^{-1}$) in Eq. (1). The same typographical error is found in ECMWF (2016), and $k_w$ is set to 0.6 – see Eq. (8.151) in ECMWF (2016). If Eq. (1) is correct, $k_w$ should be the heat diffusivity which is in the order of 10$^{-7}$ magnitude. Furthermore, if $k_w$ is the heat conductivity in Eq. (1), units on the left- and right-hand sides are not same.

Therefore, the correct equation for Eq. (1) should be:

$$T_{int} - T_{cool} = \frac{\delta}{k_w} \left( Q + f_s\, SW \right) \tag{2}$$

where $k_w = h_w \cdot \rho_w \cdot c_w$.

Besides the typographical error related to the heat conductivity/diffusivity, another error is identified in the description of δ





parameterization, which is given in Eq. (6) of Zeng and Beljaars (2005) as well as in Eq. (8.155) in ECMWF (2016), i.e.:

$$\delta = 6\left\{1+\left[\frac{-16 g a_w v_w^3}{u_{*w}^4 k_w^2 \rho_w c_w}(Q+f_s\,SW)\right]^{3/4}\right\}^{-1/3} \tag{3}$$

where $g$, $v_w$, $a_w$, and $u_{*w}$ denote the acceleration of gravity (m s$^{-2}$), kinematic viscosity of water (m$^2$ s$^{-1}$), thermal expansion
coefficient of water (K$^{-1}$), and friction velocity of water (m s$^{-1}$), respectively.

In line with Eq. (2), Eq. (3) should be as follows:

$$\delta = 6\left\{1+\left[\frac{-16 g \rho_w c_w a_w v_w^3}{u_{*w}^4 k_w^2}(Q+f_s\,SW)\right]^{3/4}\right\}^{-1/3} \tag{4}$$

Even after correcting the conductivity/diffusivity issue, units on the left- and right-hand sides in Eq. (4) are found to be different (i.e., meter $\neq$ unitless). It turned out that Eq. (4) should not be about the depth of the cool skin layer ($\delta$) but about
the Saunders constant ($\lambda$ in no dimension, analogous to the Reynolds number) – Faillar et al. (1996b). Thus, the correct expressions are:

$$\lambda = 6\left\{1+\left[\frac{-16 g \rho_w c_w a_w v_w^3}{u_{*w}^4 k_w^2}(Q+f_s\,SW)\right]^{3/4}\right\}^{-1/3} \tag{5}$$

$$\delta = \frac{v_w}{u_{*w}}\lambda \tag{6}$$

In the OML model, the stability functions for the warm layer are dependent on the model configuration, such as model
integration time and depth of warm layer, as depicted in Eq. (A8) of Appendix A. Therefore, the current stability functions should be modified to align with the revised model configuration. In this study, new stability functions have been derived using ATLAS buoy data collected at 0° 140°W from 2001 to 2020 (see Fig. S1). Furthermore, as reported in Zhang et al. (2020), an adjustment of the parameter $\lambda$ can effectively alleviate the bias in $T_{skin}$. It is because the depth of cool skin layer ($\delta$) is proportional to $\lambda$, as shown in Eq. (6), and the $T_{skin}$ is proportional to $\delta$ for the given heat supply to the cool layer.
Similarly, we have introduced a scaling factor to adjust $\lambda$, and it has been set to 0.2 through a comparison with M-AREI data from two voyages conducted in the year 2020. Detailed information concerning the stability functions and the scaling factor can be found in Appendix A.





With corrections and modifications mentioned above, the implemented stand-alone OML model is now referred to as the "revised" OML model in this study. For comprehensive description regarding the model structure and configuration is found in Appendix A.

## 4 Inputs and validation data

To simulate the diurnal variation of $T_{int}$ using the revised OML model, it is necessary to provide heat exchanges at the interface as inputs to the OML model, as outlined in Eq. (2). Instead of simulating $T_{int}$ with forecasting model, this study utilizes ERA5 surface heat fluxes as inputs to the revised model. A key benefit of employing ERA5 heat fluxes is the ability to compare our simulation results with ERA5 $T_{int}$ analysis. This comparison helps in examining and potentially correcting the abnormal features identified in Figs. 1–2 because inputs to the model should be same between ERA5 reanalysis and this calculation. Of course, the simulation results are validated by directly comparing them with data acquired from buoy-mounted thermistors and ship-board IR interferometers. The following subsections provide description on the input and validation/comparison data used in this study.

*a. ERA5 hourly ocean skin temperature ($T_{int}$) and atmospheric/oceanic forcing*

The ERA5 reanalysis, the fifth reanalysis project of ECMWF, provides high-resolution global atmospheric and surface data from 1940 onwards. This reanalysis is based on the forecast model of the integral forecast system (IFS) version in Cy41r2 and the 4D-Var data assimilation system (Hersbach et al., 2020). In ERA5, the hourly ocean skin temperature (i.e., $T_{int}$), providing diurnal features, is simulated using the OML model (ECMWF, 2016). This model accounts for interactions between the atmosphere and ocean with consideration of atmospheric forcing and oceanic conditions. In particular, the oceanic condition is represented by the ERA5 SST (i.e., $T_{fnd}$), which serves as a baseline temperature for simulating the time-varying $T_{int}$ in the OML model. The ERA5 SST is updated daily at 21 UTC with a new OSTIA level 4 SST$_{fnd}$ data and remains fixed until the next updating cycle.

In this study, ERA5 single-level surface parameters are used. As inputs for running the revised OML model, SST, surface net solar and thermal radiation fluxes, surface latent and sensible heat fluxes, and wind speed at 10 m are used. All data are given in a 0.25° × 0.25° latitude-longitude grid format with an hourly resolution. The analysis domain is limited to the open ocean area between 60°N and 60°S.

*b. M-AERI-measured skin sea surface temperature ($T_{skin}$)*

Simulated $T_{skin}$ is compared with the $T_{skin}$ measured from the marine-atmospheric emitted radiance interferometer (M-AERI), which is a Fourier transform IR interferometer specifically designed for long-term deployment on ship decks. The M-AERI





operates within the wavelength range of 3 to 18 μm with a spectral resolution of 0.5 cm⁻¹ and an incidence angle of 55° and provides $T_{skin}$ and near-surface air temperature. The details of the sensor and the retrieval algorithm are found in Minnett et al. (2001).

For the validation using M-AERI measurements, we use $T_{skin}$ at a depth of approximately 10 μm with a temporal resolution of 5 minutes. The data are available over an 8-year period from 2013 to 2020, taken from 24 voyages (Minnett et al., 2020). Among the 24 voyage data, 2 voyages in 2020 are used for determining the scaling factor for parameter λ, and remaining 22 voyages from 2013 to 2019 are used for model validation. For detailed information about the individual cruises/vessels and their tracks, see Fig. S2 and Table S1. Among the quality-controlled data, we only select measurements with an uncertainty range within 0.1 K (Minnett et al., 2005) and at least 25 km away from the coastline. Subsequently, the data are thinned to match the model's one-hour temporal resolution. A spatial collocation procedure is also performed, whereby data points are mapped to the nearest model grid point.

*c. ATLAS buoy-measured bulk sea surface temperature ($T_{1m}$)*

The automated temperature line acquisition system (ATLAS) mooring buoys, deployed within the tropical atmosphere ocean (TAO) array, play a crucial role in the global climate observing system. These buoys are equipped with both atmospheric and oceanic sensors. The atmospheric sensors measure various variables, including rain rate, shortwave and longwave radiation fluxes, air temperature, relative humidity, pressure, and wind. The oceanic sensors primarily consist of 11 temperature thermistors positioned at depths ranging from 1 to 500 m, enabling temperature sounding throughout the ocean column up to 500 m depth (Miburn et al., 1996).

In this study, we use temperature data collected at a depth of 1 m (i.e., $T_{1m}$) with a temporal resolution of 10 minutes from 54 buoys over a 20-year period from 2001 to 2020. Among the 54 buoy data, a single buoy located at 0° 140°W is used for modifying model parameterizations, and remaining 53 buoys are used for model validation. The geographical distributions of buoys and their coordinates are found in Fig. S2 and Table S2. We only select measurements that are continuously observed throughout 24 hours and whose diurnal variations fall within a 3-standard deviation (1.9 K) for the overall buoy's diurnal variations. The data are thinned at one-hour intervals to match the model's temporal resolution. Since all of the buoys were deployed at points that coincided with the model grid points, there is no need for the additional spatial collocation procedure.

## 5 Results

### 5.1 Validation of $T_{int}$ simulations with the revised OML model

For the simulation of hourly $T_{int}$, we use the revised stand-alone OML model with atmospheric forcing and $T_{fnd}$ provided by




ERA5 as inputs. It must be ideal to validate the $T_{int}$ simulations against observations, but $T_{int}$ observations are not currently

available. Thus, instead of referring to $T_{int}$ observations, we indirectly validate by deriving $T_{skin}$ and $T_{1m}$ from simulations and comparing them against observations. In the OML model, the temperature profile between $T_{int}$ and $T_{cool}$ within the cool skin layer is assumed to be linear. Then, $T_{skin}$ at a depth of 10 μm can be calculated with $T_{int}$ and $T_{cool}$:

$$T_{skin} = \frac{\delta - 10^{-5}}{\delta} T_{int} + \frac{10^{-5}}{\delta} T_{cool} \qquad (7)$$

The model-derived $T_{skin}$ is compared with the M-AREI-measured $T_{skin}$ from 22 voyages spanning 7 years from 2013 to 2019.

The comparison results are exhibited in Fig. 3 with statistics. The comparison shows a good agreement with a correlation coefficient of 0.99, a mean deviation of −0.08 K, and a standard deviation of 0.49 K for a total of 32,647 data points.

Validation is also made against buoy measurements. With the pre-established shape function for the temperature profile within the warm layer (see Appendix A for details), we can estimate $T_{1m}$ with $T_{cool}$ and $T_{warm}$ as follows:

$$T_{1m} = T_{cool} - \left(\frac{-1+\delta}{-5+\delta}\right)^{0.3} \left(T_{cool} - T_{warm}\right) \qquad (8)$$

We compare the model-derived $T_{1m}$ against the ATLAS buoy-measured $T_{1m}$ from 53 buoys spanning a 20-year period from 2001 to 2020. The comparison result illustrated in a scatterplot (Fig. 4) shows again a good agreement with a correlation coefficient of 0.99, a mean deviation of −0.07 K, and a standard deviation of 0.28 K, based on the use of a total 6,241,008 data points.

The good agreements shown in both comparisons ensure that the revised OML model is capable of successfully reproducing

hourly-observed ocean temperatures within ranges suggested by associated statistics. Thus, although direct observational validation of $T_{int}$ is not available, the agreement suggests that the predicted $T_{int}$ possesses the same degree of accuracy as noted in the estimated $T_{skin}$ and $T_{1m}$. It is because $T_{skin}$ and $T_{1m}$ were interpolated from predicted $T_{int}$, $T_{cool}$, and $T_{warm}$, as described in Eqs. (7) and (8).

**5.2 Diurnal variation of simulated $T_{int}$**

In order to examine whether abnormal features found in ERA5 $T_{int}$ exist in the simulations, hourly variations of simulated $T_{int}$ are given in Fig. 5 in a 3-hour interval over the same period from 12 UTC on 1 January 2020 to 21 UTC on 2 January 2020, as in Fig. 1. Furthermore, the animation depicting the hourly variation of $T_{int}$ from 1 to 5 January 2020 has been separately uploaded as a supplementary material. It is clear that the temporal disruptions noted in ERA5 at 22 UTC and 10 UTC are no longer present. The variations between the two time zones appear to be continuous and smooth without



disruption. Moreover, the spatially variable and localized patterns in ERA5 over the period from 22 UTC to 09 UTC (Fig. 1e−1h) do not exist in the simulations. Thus, it is presumed that diurnal variations of ERA5 $T_{int}$ are problematic, as two different alternating regimes every 12 hours should not be reasonable. We cannot attribute any atmospheric or oceanic forcings that would give rise to the periodic oscillations in the $T_{int}$ distribution with a 12-hour cycle. By contrast, in our simulation, high-variation magnitude areas continuously move westward and circulate over the globe, presumably following 255 a 24-hour cycle coinciding with solar forcing.

**5.3 Comparison against surface net solar radiation**

Given that solar radiation is a major driving force for the $T_{int}$ variations during the daytime (Kawai and Yada, 2007), there should be a strong correlation between $T_{int}$ and solar radiation during the day. Thus, the comparison of $T_{int}$ with the surface net solar radiation flux can be used as another means to examine the model performance. Fig. 6 illustrates the geographical 260 distributions of $T_{int}$ departures from $T_{fnd}$ (i.e., $T_{int} - T_{fnd}$) for our simulation and ERA5 at 00 UTC and 12 UTC on January 2, 2020, along with the ERA5 surface net solar radiation flux. The comparison for this study (Fig. 6a vs. 6c and Fig. 6b vs. 6d) clearly shows that high solar radiation areas centered at 180° and at 0° are well correlated with the positive $T_{int}$ departure areas. Nighttime areas are found to be in small negative departure areas, consistent with the expectation that the $T_{int}$ variation is rather small during the nighttime (Gentemann et al., 2003). On the other hand, ERA5 at 00 UTC (Fig. 6a vs. Fig. 6e) has 265 no clear correspondence to the solar radiation over the Pacific Ocean, and does not show day-and-night contrast. However, 12 hours later (at 12 UTC), ERA5 shows a slightly improved match with the distribution of solar radiation, but it still shows a weaker correlation over the Atlantic Ocean (Fig. 6b vs. Fig. 6f). Interestingly, this time, the nighttime distribution over the Pacific Ocean appears to be quite similar to one found in our simulation, different from the features noted 12 hours ago (at 00 UTC). These results draw a conclusion that the ERA5 $T_{int}$ variation is not coherent or modulated by surface net solar 270 radiation. This seems true at least during a 12-hour period from 22 UTC to 09 UTC.

Furthermore, we compare the time series of $T_{int}$ between this study and ERA5 for a week from 1−7 January 2020 at four locations (i.e., 20°S 80°E, 20°N 180°, 0° 155°W, and 30°S 120°W) in Fig. 7. Also presented together is the time series of surface net solar radiation flux over the same period. It is clear that the revised OML model produces $T_{int}$ variations that are well correlated with the solar radiation. On the contrary, the ERA5 time series displays no clear correlation with solar 275 radiation, whereas ERA5 $T_{int}$ values consistently exhibit lower temperatures than those obtained from our simulations. These characteristics appear to extend beyond this specific analysis period. Upon a comprehensive examination of the entire year 2020 at 0° 155°W, it becomes evident that these traits persist throughout all seasons (Fig. S3), underscoring that ERA5 daily mean values are predominantly lower than those indicated by simulations. In terms of seasonal averages, ERA5 shows temperatures 0.13–0.15 K cooler than the simulations at this particular equatorial Pacific location.



## 6 Conclusions

Motivated by unrealistic behaviors noted in ERA5 $T_{int}$ data, we revisited the OML model, being utilized for predicting $T_{int}$ at ERA5. After identifying typographical errors in the literature describing the OML model, we rectified them and validated the model-simulated temperature profiles in the upper ocean boundary layer. This validation was against observations from ship-board IR interferometers and buoy-mounted thermistors, which represent $T_{skin}$ and $T_{1m}$, respectively. The comparison demonstrated good agreement, with an absolute mean deviation smaller than 0.1 K and a standard deviation below 0.5 K, leading to a suggestion that $T_{int}$ possesses the same degree of accuracy. In addition, the revised OML model also indicates a strong correlation between diurnal variations of simulated $T_{int}$ and solar radiation in terms of both geographical distributions and time series.

Given that the revised OML model employs the same underlying physics as in the ERA5 OML model and that the simulations used identical inputs as for ERA5, we anticipated a high degree of similarity between the two $T_{int}$ datasets. However, the diurnal variations of ERA5 $T_{int}$ appear to deviate significantly from scientifically sound expectations when compared against model-simulated $T_{int}$ and surface net solar radiation flux. In particular, it shows little correlation with solar radiation, despite solar radiation being a major driving force for SST variations during the daytime. These are characteristic features persisting throughout the year 2020, which was the period we analyzed as a case in point, suggesting that results are not fortuitous. In addition to the unreasonable features such as 12-hour alternating spatial patterns found in ERA5, daily averages substantially cooler than our simulations and unorganized relationship with solar radiation inevitably lead to the conclusion that ERA5 may not accurately represent the nature of diurnal variations in $T_{int}$. Therefore, this study demonstrates that the OML model, which we revised, is capable of producing $T_{int}$ diurnal variations comparable to observations. It holds the potential to be a valuable tool for generating accurate $T_{int}$ diurnal variations.

## Appendix A

In this study, the OML model was revised by examining the theoretical development made by Fairall et al. (1996b) and Zeng and Beljaars (2005). Some typographical errors noted in Zeng and Beljaars (2005) were corrected as described in the main text. During the model revision, stability functions for the warm layer were updated, and the scaling factor for the Saunders constant was introduced. The required input parameters to the model are the surface net solar radiation flux (SW), surface net thermal radiation flux (LW), latent heat flux (E), sensible heat flux (H), neutral wind speed at 10 m height in the atmosphere ($u_{10m}$), and foundation SST ($T_{warm}$). The OML model also incorporates the prescribed state parameters summarized in Table S3. The shape function of the temperature profile within the warm layer is expressed as follows:





$$T_z = T_{cool} - \left(\frac{-z+\delta}{-d+\delta}\right)^v (T_{cool} - T_{warm}) \tag{A1}$$

where z is the depth, ν is the shape parameter of the warm layer temperature profile, and δ and d (both in meters) denote
depths of cool and warm layers. Thus, within a 1-D column, the warm layer vertically extends from δ to d, and $T_{cool}$, $T_z$, and
$T_{warm}$ (all in K) refer to the temperatures at depths of δ, z, and d, respectively.

The thermal expansion coefficient of water ($\alpha_w$ in $K^{-1}$) is parameterized as follows (ECMWF, 2016):

$$a_w = 10^{-4} \cdot \max[T_{warm} - 273,\ 1] \tag{A2}$$

The friction velocity in the air ($u_{*a}$ in m $s^{-1}$) is derived from the neutral wind speed at 10 m ($u_{10m}$) using Eq. (A3), as
introduced by Andreas et al. (2012).

$$u_{*a} = 0.239 + 0.0433 \left[ u_{10m} - 8.271 + \sqrt{0.12(u_{10m} - 8.271)^2 + 0.181} \right] \tag{A3}$$

Subsequently, the friction velocity in water ($u_{*w}$ in m $s^{-1}$) is calculated as follows:

$$u_{*w} = u_{*a} \sqrt{\frac{\rho_a}{\rho_w}} \tag{A4}$$

We define Q = H + E + LW because of their cooling effects. It is noted that the convention for vertical fluxes is established
as positive downwards and negative upwards.

Equation A5 describes how $T_{cool}$ evolves over time:

$$\frac{\partial \Delta T}{\partial t} = \frac{\alpha}{d}[Q + f_d\ SW] - \frac{\beta}{d}\frac{u_{*w}}{\phi}\Delta T \tag{A5}$$

where ΔT is the difference between $T_{cool}$ and $T_{warm}$ (i.e., $T_{cool} - T_{warm}$), and t is time. The bulk coefficients α (in K $m^3$ $J^{-1}$) and
β (dimensionless) are defined as follows:

$$\alpha = \frac{v+1}{v c_w \rho_w} \tag{A6}$$

$$\beta = k(v+1) \tag{A7}$$




In Equation A5, the first term on the right-hand side represents the heat flux from energy exchange with the atmosphere, while the second term on the right-hand side is for the ocean internal heat transport due to the turbulent diffusion.

The numerical solution for Eq. (A5) can be obtained as follows:


$$T_{\text{cool}}^{(n+1)} = T_{\text{warm}}^{(n)} + \frac{\Delta t}{d}\left\{\alpha\left[Q^{(n)} + f_d\ SW^{(n)}\right] - \beta\frac{u_{*w}^{(n)}}{\phi^{(n)}}\left(T_{\text{cool}}^{(n)} - T_{\text{warm}}^{(n)}\right)\right\} \qquad (A8)$$

where the superscripts (n) and (n+1) indicate (n)th and (n+1)th time step. As $T_{\text{warm}}$ is fixed in time, $T_{\text{cool}}$ at (n+1)th time step can be determined by explicitly considering the variation induced by atmospheric forcing at the (n)th time step during the temperature deviation ($\Delta T$).

The shortwave absorptivity in the warm layer ($f_d$ in dimensionless unit) is defined in Eq. (A9), with the group coefficients
[A] and [B] that are defined as [$A_1, A_2, A_3$] = [0.28, 0.27, 0.45] and [$B_1, B_2, B_3$] = [71.5, 2.8, 0.07].

$$f_d = 1 - \sum_{i=1}^{3} A_i e^{-d \cdot B_i} \qquad (A9)$$

The stability functions ($\phi$ in dimensionless unit) of the warm layer are defined by Eqs. (A10), (A11), and (A15), which are functions of stability parameter $\zeta$ (= d / L). Here, we introduce new empirical $\phi$-$\zeta$ relations specific to the revised model configuration. These relations were formulated based on the use of ATLAS buoy measurements at 0° and 140°W from 2001
to 2020. The scatterplots of buoy-derived $\zeta$ vs. $\phi$ are presented in Fig. S4, which are similar to the relation proposed by Stiperski and Calaf (2023). Derived stability functions are:

$$\phi = 2 + 12\zeta^{-0.5} \qquad (\zeta > 0) \qquad (A10)$$

$$\phi = 2 + 8(-\zeta)^{-0.5} \qquad (\zeta < 0) \qquad (A11)$$

Here, L (in meters) represents the Obukhov length, which is positive (negative) for stable (unstable) stratification, and it is
given as follows:

$$L = \frac{u_{*w}^3}{kF_d} \qquad (A12)$$

where $F_d$ is the buoyancy flux (in $m^2\ s^{-3}$), and is given by:





$$F_d = \frac{g\alpha_w}{\rho_w c_w}\left(Q + f_d\ SW\right)$$ (A13)

In addition, to account for the phenomena of relatively warmer temperature persisting for a certain duration after sunset,
when $T_{cool}$ is warmer than $T_{warm}$ (i.e., $T_{cool} - T_{warm} \geq 0$), and simultaneously, the net flux at the air-sea interface is negative
(i.e., $Q + f_d\ SW < 0$), the numerical solution for $T_{cool}$ is replaced as follows:

$$T_{cool}^{(n+1)} = T_{warm}^{(n)} + \frac{\Delta t}{d}\left[-\beta\frac{u_{*w}^{(n)}}{\phi^{(n)}}\left(T_{cool}^{(n)} - T_{warm}^{(n)}\right)\right]$$ (A14)

For the case of such sunset duration, the associated stability function and buoyancy flux within the warm layer are separately
defined as follows:


$$\phi = 2 + 4\zeta^{-1}$$ (A15)

$$F_d = u_{*w}^2\sqrt{\frac{vg\alpha_w}{5d}\left(T_{cool} - T_{warm}\right)}$$ (A16)

At this point, $T_{cool}$ at the (n+1)th time step becomes obtainable. Moreover, if necessary, the temperature at a depth of z within
the warm layer, $T_z$, can be determined using Eq. (A1) from $T_{cool}$ and $T_{warm}$ at the (n+1)th time step.

The boundary condition at the air-sea interface satisfies the energy equilibrium, which is expressed as follows:


$$k_w\frac{\partial T}{\partial z} = Q + f_s\ SW$$ (A17)

The numerical solution for $T_{int}$ (in K) at the (n+1)th time step is defined based on the implicit approach:

$$T_{int}^{(n+1)} = T_{cool}^{(n+1)} + \frac{\delta^{(n)}}{k_w}\left(Q^{(n)} + f_s^{(n)}\ SW^{(n)}\right)$$ (A18)

where $f_s$ (dimensionless) is the shortwave absorptivity of the ocean surface,

$$f_s = 0.065 + 11\delta - \frac{6.6\times10^{-5}}{\delta}\left(1 - e^{-\delta/8\times10^{-4}}\right)$$ (A19)



and the depth of the cool skin layer ($\delta$) is defined,

$$\delta = \frac{v_W}{u_{*W}} \lambda \qquad (A20)$$

In Eq. (A20), $\lambda$ (dimensionless) denotes the Saunders constant as below:

$$\lambda = 6 \left\{ 1 + \left[ \frac{-16 g \rho_w c_w a_w v_w^3}{u_w^4 k_w^2} \left( Q + f_s\ SW \right) \right]^{\frac{3}{4}} \right\}^{-\frac{1}{3}} \qquad (A21)$$

It should be noted that Eqs. (A19) and (A21) are interdependent, i.e., $f_s = f(\lambda)$ and $\lambda = f(f_s)$. Thus, to obtain the numerical
solutions for both variables, an iterative minimum residual approach is needed. Fairall et al. (1996b) proposed a slightly
different parameterization for $\lambda$, i.e.:

$$\lambda = 6 \left[ 1 + \left( \frac{-16 g \rho_w c_w a_w v_w^3}{u_w^4 k_w^2} Q \right)^{\frac{3}{4}} \right]^{-\frac{1}{3}} \qquad (A22)$$

Given that the discrepancy between Eq. (A21) and Eq. (A22) emerges at the nanometer scale within the depth $\delta$, the
difference can be regarded as negligible. In this study, Eq. (A22) was chosen for $\lambda$.

The Saunders constant $\lambda$, analogous to the Reynolds number, plays a crucial role in simulating $T_{int}$ by determining the depth
of the cool skin layer, $\delta$. Zhang et al. (2020) reported that the bias of $T_{skin}$ can be effectively mitigated by adjusting $\lambda$.
Similarly, we introduced a scaling factor to adjust $\lambda$, and it is set to 0.2, determined from the comparison with M-AREI data
from the year 2020. This scaling factor yielded a reduction in bias from $-0.38$ to $-0.12$, and the resultant scatterplot of
applying it to the 2020 data is presented in Fig. S4. As noted by Saunders et al. (1967), the magnitude of $\lambda$ is in a zeroth
order, implying the presence of laminar flow within the cool skin layer. The scaled $\lambda$ still falls within the magnitude of the
same zeroth order, thereby retaining its laminar flow characteristics.

**Data Availability**

The ERA5 hourly reanalysis data on single levels is accessible for download through the Climate Data Store (CDS), which is
implemented by ECMWF as a part of the Copernicus Climate Change Service (C3S)
(https://cds.climate.copernicus.eu/cdsapp#!/dataset/reanalysis-era5-single-levels?tab=form). The dataset for ship-based high-
resolution SST$_{skin}$ from M-AERI, spanning the years 2013 to 2020, is available under open access from the repository of the



University of Miami Libraries (https://doi.org/10.17604/bswq-0119). The dataset for the ATLAS mooring buoy-measured $SST_{1m}$ within the TAO array can be obtained from the website of the National Data Buoy Center (NDBC) in the National Oceanic and Atmospheric Administration (NOAA) (https://www.ndbc.noaa.gov).

**Code Availability**

All code generated in this study can be provided by the corresponding authors upon request.

**Author contribution**

**EJK**: Conceptualization, Formal analysis, Investigation, Methodology, Software, Validation, Visualization, Writing – original draft, Writing – review and editing; **BJS**: Conceptualization, Methodology, Writing – original draft, Writing – review and editing, Funding acquisition, Supervision; **SWK**; Writing – review and editing, Resources, Funding acquisition; **WK**: Software, Writing – review and editing; **YCK, SBK, HWC, CL**: Writing – review and editing

**Competing interests**

The authors declare that they have no conflict of interest.

**Acknowledgements**

This work was supported by the Basic Science Research Program through the National Research Foundation of Korea (NRF) funded by the Ministry of Education (No. RS-2023-00271704), the Development of Numerical Weather Prediction and Data Application Techniques funded by the Korea Meteorological Administration (KMA) (No. KMA2018-00721), and the National Research Foundation of Korea (NRF) grant funded by the Korean government (MSIT) (No. NRF-2021R1A4A5032320).

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





**List of figures**

**Figure 1.** Geographical distributions of hourly variations in ERA5 $T_{int}$ from 12 UTC 1 January to 21 UTC 2 January in 2020 with a 3-hour interval. Values are given in departures of ERA5 $T_{int}$ from ERA5 $T_{fnd}$ (i.e., $T_{int} - T_{fnd}$).

**Figure 2.** Same as Figure 1, but given in hourly departures at (a) 09 UTC, (b) 10 UTC, (c) 21 UTC, and (d) 22 UTC 1 January 2020.

**Figure 3.** Scatterplots of model-derived $T_{skin}$ vs. M-AREI-measured $T_{skin}$ over 7 years from 2013 to 2019. The color scale is for the data frequency. The 'r', Bias, and Std. Dev. in the inlet denote correlation coefficient, mean deviation, and one standard deviation, respectively.

**Figure 4.** Same as in Figure 4 except for model-derived $T_{1m}$ vs. ATLAS buoy-observed $T_{1m}$ from 53 ATLAS buoys from 2001 to 2020.

**Figure 5.** Same as in Figure 1 but for $T_{int}$ simulated with the revised OML model.

**Figure 6**. Geographical distributions of (a–b) surface net solar radiation flux from ERA5 and hourly variations of $T_{int}$ from (c–d) the revised OML model simulations and (e–f) ERA5 at 00 UTC (left panels) and 12 UTC (right panels) on 2 January 2020.

**Figure 7.** Time series of $T_{int}$ from the model simulations and ERA5 (top panel of each figure), and ERA5 surface net solar radiation flux (bottom panel of each figure), for a week period from January 1 to 7, 2020, at four locations: (a) 20°S 80°E, (b) 20°N 180°, (c) 0° 155°W, and (d) 30°S 120°W.





**Figure 1.** Geographical distributions of hourly variations in ERA5 $T_{int}$ from 12 UTC 1 January to 21 UTC 2 January in 2020 with a 3-hour interval. Values are given in departures of ERA5 $T_{int}$ from ERA5 $T_{fnd}$ (i.e., $T_{int} - T_{fnd}$).





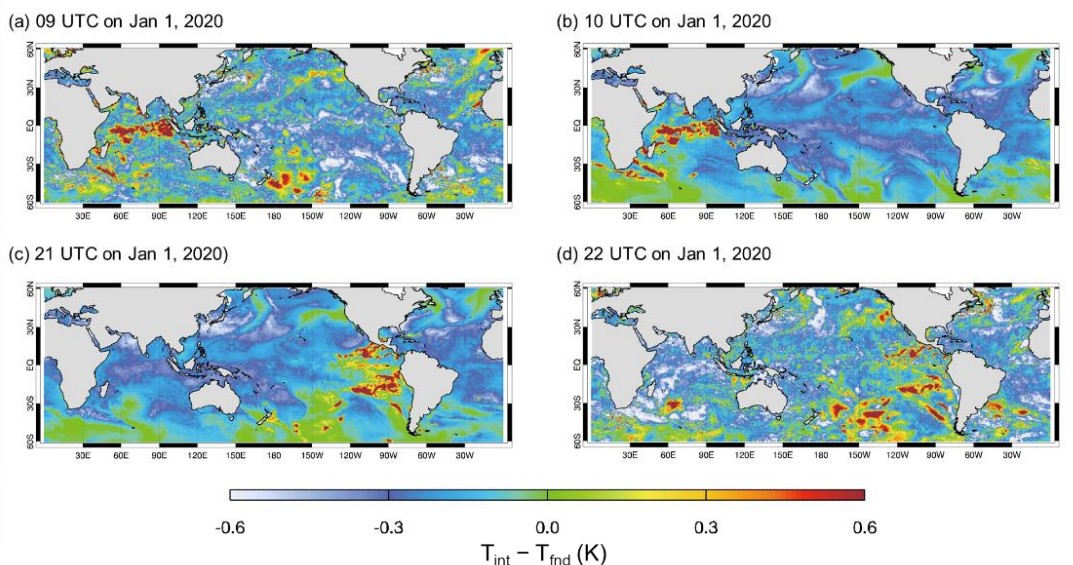

**Figure 2.** Same as Figure 1, but given in hourly departures at (a) 09 UTC, (b) 10 UTC, (c) 21 UTC, and (d) 22 UTC 1 January 2020.

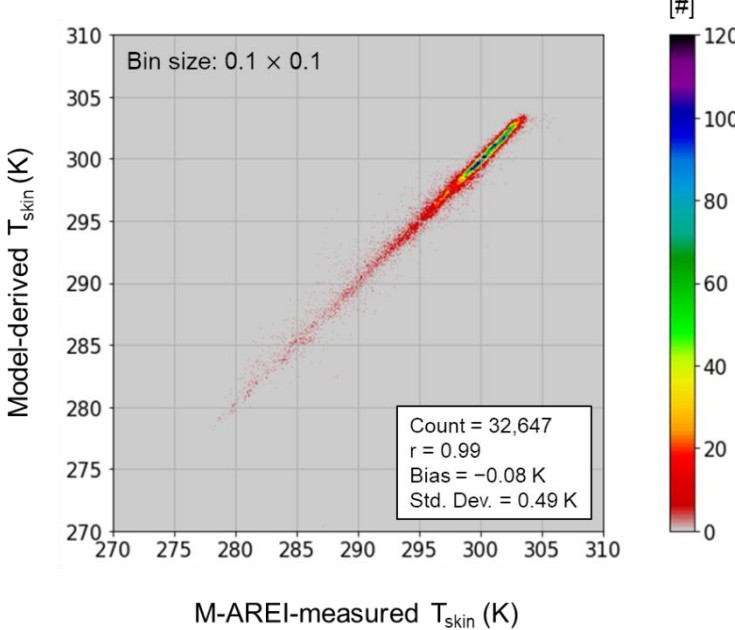

**Figure 3.** Scatterplots of model-derived $T_{skin}$ vs. M-AREI-measured $T_{skin}$ over 7 years from 2013 to 2019. The color scale is for the data frequency. The 'r', Bias, and Std. Dev. in the inlet denote correlation coefficient, mean deviation, and one standard deviation, respectively.



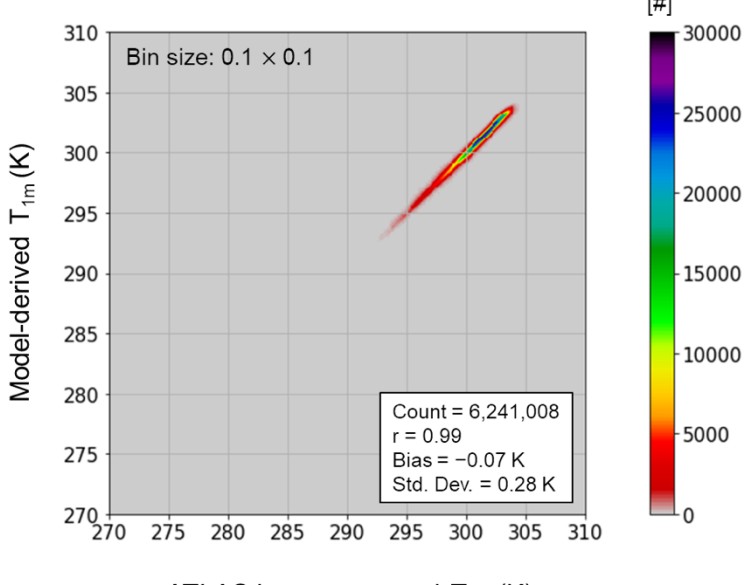

**Figure 4.** Same as in Figure 4 except for model-derived $T_{1m}$ vs. ATLAS buoy-observed $T_{1m}$ from 53 ATLAS buoys from 2001 to 2020.

**Figure 5.** Same as in Figure 1 but for $T_{int}$ simulated with the revised OML model.

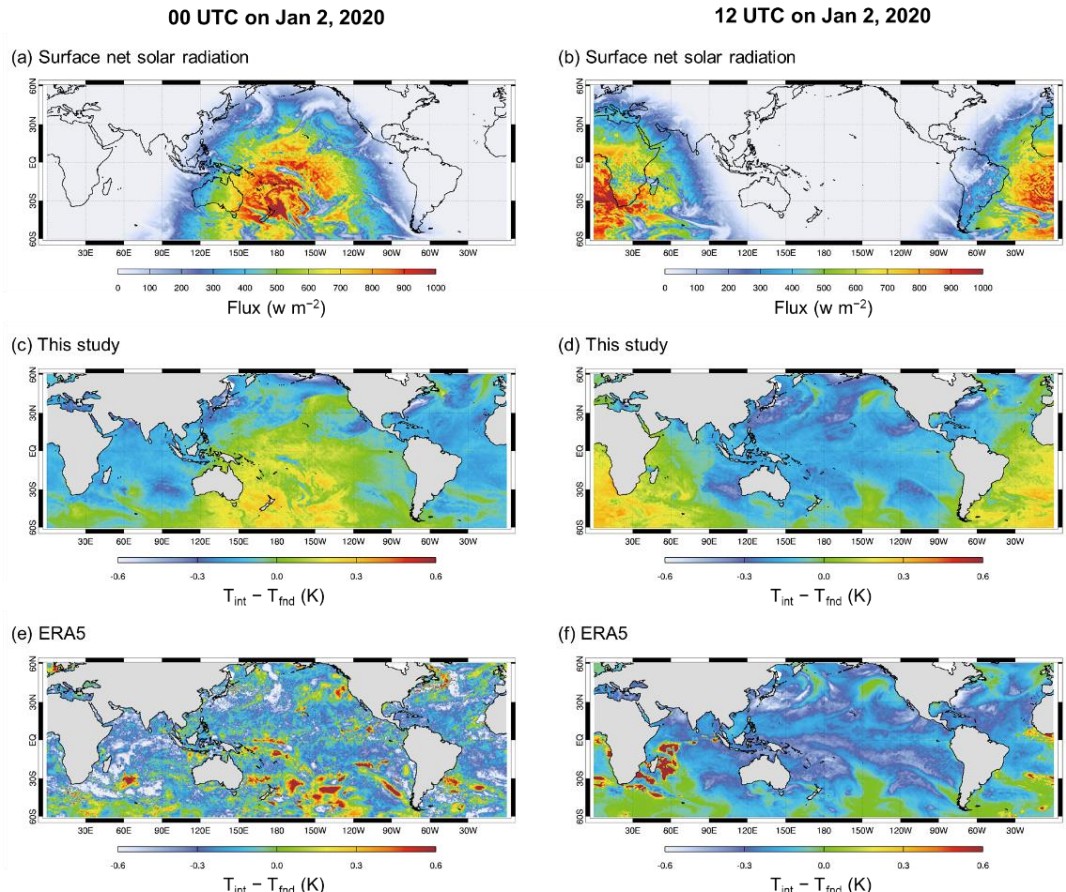

**Figure 6**. Geographical distributions of (a–b) surface net solar radiation flux from ERA5 and hourly variations of $T_{int}$ from
(c–d) the revised OML model simulations and (e–f) ERA5 at 00 UTC (left panels) and 12 UTC (right panels) on 2 January
2020.



**Figure 7.** Time series of $T_{int}$ from the model simulations and ERA5 (top panel of each figure), and ERA5 surface net solar radiation flux (bottom panel of each figure), for a week period from January 1 to 7, 2020, at four locations: (a) 20°S 80°E, (b) 20°N 180°, (c) 0° 155°W, and (d) 30°S 120°W.