# Peer review of "A revised ocean mixed layer model for better simulating the diurnal variation of ocean skin temperature"

_Geoscientific Model Development, 2024_

## Author Comment (AC2)

**Response to Referee 1**

*This work identifies several errors in the definition of ocean skin temperatures in the ECMWF ocean mixed layer model (OML). The authors propose a modified OML model that corrects for these errors, and evaluate its performance compared with the previous model. Although the methods and results presented in the paper suggest that this new model improves on its predecessor, which is indeed an important result and worth reporting, the layout of the manuscript and the diagnostics used here are not sufficiently convincing. I recommend resubmitting after addressing the following comments.*

We appreciate your comments on our manuscript, which have been instrumental in guiding our revisions. In this revision, we tried hard to meet your expectations and address all the concerns raised. Detailed responses to each of your points are as follows.

**Major comments**

*1. The language in the manuscript needs a major rewrite. I've included a few comments below but perhaps professional editing will be helpful.*

Thank you for your suggestion. We have revised the manuscript, and it has been professionally edited by a native speaker. We believe it is now in much better shape.

*2. There are many acronyms defined but rarely used, and it is distracting from those variables that are important for the work presented here. I suggest emphasizing which variables are the most relevant and try to minimize the use of other acronyms and definitions throughout.*

We fully agree with your suggestion. In the revised version, to maintain the natural flow of the paper and avoid confusion, we have removed acronyms and definitions that are unnecessary or distracting, minimizing their use to include only those that are essential.

*3. On that note, I found the explanation of the various SST definitions very confusing, and in particular how come only the upper 5 meters or so are affected by the diurnal cycle where we should also expect mixing effects (due to surface forcing) to extend far below that. Please rewrite this while taking into account other OML models and how they treat the vertical profile of temperature (you have some discussion on this in lines 117-126 but it contradicts your original assumption that SST does not change below 5-10 m).*

SST definitions

Recognizing that the various definitions and abbreviations of SST may have caused confusion, we have removed subsection 2a and simplified the overall terminology in the revised manuscript. Instead, only the relevant temperatures are defined at the point of use. Furthermore, we have included a new figure (Fig. 1 in the revised version) to visually present the terms, along with a brief description of the revised OML model, to enhance understanding of the OML

model structure and SST definitions.

[Figure]

**Figure 1.** Schematic diagram illustrating the ocean mixed layer (OML) model. $T_s$, $T_{cool}$, and $T_{warm}$ represent the temperatures at three levels of the OML model. In ERA5, skin temperature and SST correspond to $T_s$ and $T_{warm}$, respectively. Temperature $T_z$ represents the temperature at depth z. The thick solid lines depict the temperature profiles as expressed by two equations in the cool skin and warm layers.

Top 5 m as an OML depth

The warm layer is defined as a region in the upper few meters of the ocean where solar radiation causes significant warming relative to the deep mixed layer temperature (Fairall et al., 1996). The bottom of the warm layer is not where the diurnal variation of temperature vanishes but where diurnal variation becomes less meaningful. Consequently, the depth of the warm layer can vary, as noted in various studies: 2–4 meters (Zeng and Beljaars, 2005; ECMWF, 2016), 5–10 meters (Fujiwara et al., 2017), 1–5 meters (GHRSST Science Team, 2022). In our study, we use a depth of 5 meters, the upper bound suggested by the GHRSST Science Team (2022), as the depth of the OML.

In the revised version, we have newly added the following phrases:

"The warm layer is defined as the upper few meters of the ocean where solar radiation causes significant warming relative to the deep mixed layer temperature (Fairall et al., 1996b). The

bottom of the warm layer is not where the diurnal variation of temperature vanishes but where this variation becomes less meaningful. Consequently, the depth of the warm layer can vary, as noted in various studies: 2–4 meters (Zeng and Beljaars, 2005; ECMWF, 2016), 5–10 meters (Fujiwara et al., 2017), 1–5 meters (GHRSST Science Team, 2022). In this study, we have adopted a depth of 5 meters, the upper bound proposed by the GHRSST Science Team (2022), as the depth of the warm layer.

*4. The main motivation, as presented in the manuscript, is due to inconsistent patterns in $T_{int}$ given by ECMWF (Figs 1-2), however it is hard to judge just based on the text that these patterns are sufficiently wrong. It would be helpful to show a more quantitative metric (rather than the present qualitative one) that motivates the error. If the main motivation is the error in equation 1 then you should start with that and then use the plots to demonstrate further.*

We admit that this work was initiated upon finding that ERA5 skin temperature ($T_s$) exhibits erroneous features, specifically changing spatial patterns at 10 UTC and 22 UTC. Subsequently, we identified errors in the prognostic scheme employed in the ECMWF model. Due to this, our initial presentation focused on the erroneous features before detailing our corrective measures, which may have seemed awkward.

Following your suggestion, we have restructured the manuscript for better reading. We now report the model errors and the subsequent revisions first, and then describe the erroneous features in the ERA5 data. To quantitatively measure the spatially varying patterns at 10 UTC and 22 UTC, we present the pattern correlation coefficient between two consecutive distributions of ERA5 $T_s$ as a time series (now in new Fig. 6).

[Figure]

**Figure 6.** Time series of the pattern correlation coefficient for ERA5 $T_s$ variations over the global ocean between two consecutive hourly distributions (between nth and n+1st hourly data, expressed as n–n+1 UTC), in the period from January 1 to January 5, 2020. Only 9–10 UTC and 21–22 UTC are given as labels.

In Fig. 6, the pattern correlation coefficients remain above 0.97 except for the 09–10 UTC and 21–22 UTC periods, which show sudden drops to approximately 0.8. This indicates significant

pattern transitions at 10 UTC and 22 UTC. The regularity and repeatability of these correlation dips suggest that this phenomenon is systematic and recurs at specific times each day. These findings support our assertion that $T_s$ variations exhibit spatial disruptions at 09–10 UTC and 21–22 UTC.

Accordingly, in the revised version, we have newly added the following phrases, together with Fig. 6:

"The pattern transition at 09 UTC and 22 UTC can be quantitatively assessed using the pattern correlation coefficient between two consecutive hourly $T_s$ distributions. This coefficient provides an index of the similarity or difference between the two maps, based on the notion that temperature variations should not change abruptly within one hour. The pattern correlation coefficient is calculated using consecutive hourly $T_s$ distributions over the global ocean from January 1 to 5, 2020. The resulting time series is presented in Fig. 6. The time series indicates that the coefficient, which remains above 0.97, suddenly drops to approximately 0.8 at 09–10 UTC and 21–22 UTC. This abrupt and significant drop indicates a notable change in the spatial pattern at 10 and 22 UTC, while a similar pattern is maintained over the following 12 hours once the pattern changed."

*5. $k_w$ in Eq. 1 is redefined to be the heat diffusivity on the order of $10^{-7}$, instead of thermal conductivity in the original equation, which was on the order of 0.6. These are many orders of magnitude that have likely affected the model performance and stability. I think it is important to add some discussion on this issue*

The original equation is as follows:

$$T_s - T_{\text{cool}} = \frac{\delta}{k_w \rho_w c_w} \left( H + E + LW + f_s\, SW \right) \qquad (1)$$

The confusion arises from the incorrect use of $k_w$ (thermal conductivity, 0.6 W m$^{-1}$ K$^{-1}$) instead of use of thermal diffusivity (~$1.40 \times 10^{-7}$ m² s$^{-1}$). This causes a unit inconsistency between the left and right sides of the equation. Defining $k_w$ as thermal diffusivity would resolve the problem. However, correcting the equation in this study, we follow the convention that $k_w$ is normally designated as thermal conductivity while $h_w$ is designated as thermal diffusivity. Then, the corrected version of the equation should then be:

$$T_s - T_{\text{cool}} = \frac{\delta}{h_w \rho_w c_w} \left( Q + f_s\, SW \right) \qquad (2)$$

where $k_w = h_w \cdot \rho_w \cdot c_w$.

Although the units differ on both sides of Eq. (1), we can examine the impact of using $k_w = 0.6$ instead of $1.40 \times 10^{-7}$. When $k_w = 0.6$ is used, the denominator $k_w \cdot \rho \cdot c_w$ in Eq. (1) becomes roughly $4 \times 10^6$ times larger than when using $k_w = 1.40 \times 10^{-7}$. Additionally, the incorrect assignment of thermal diffusivity affects the depth of the cool skin layer ($\delta$) in the numerator in Eq. (1). Our

calculations indicate that the wrong coefficient causes $\delta$ to be approximately $1\times10^{-4}$–$3\times10^{-4}$ times that of the correct model -- see #6. Overall, the incorrect assignment induces "$T_s - T_{cool}$" to be about $10^{-3}$ of the expected values, approaching near zero (i.e. $T_s \approx T_{cool}$) regardless of the heating magnitudes at the surface, and resulting in the temperature variability of $T_s$ being nearly the same as that of $T_{cool}$. Considering that the largest variability occurs at the ocean surface, the diurnal variation of $T_s$ should be smaller. This interpretation is found consistent with the much smaller diurnal variation of ERA5 shown in Fig. 9.

In the revised version, we have modified the relevant sentences as follows:

"In Eq. (1), $k_w$ should be the heat diffusivity, which is on the order of $10^{-7}$. Even if we ignore the unit discrepancy, the temperature difference derived using $k_w = 0.6$ is roughly $10^3$ times smaller than that obtained using correct heat diffusivity, resulting in a near-zero temperature difference between top and bottom of cool skin layer."

*6. It appears that there is some error in the transition between Eqs. 3 and 4 after the $k_w$ is redefined. Because $k_w = h_w(\rho_w c_w)^{-1} \rightarrow k_w^2 = h_w^2(\rho_w c_w)^{-2}$. However, in Eq. 3: $(\rho_w c_w)^{-1}$ are not squared, which is either a typo or not an equality with Eq. 4.*

Here again, we have to clarify wrongly defined heat conductivity in Eq. (3), which should be heat diffusivity. Following the correction of $k_w \rightarrow h_w$, Eq. (3) should become (3a).

$$\delta = 6\left\{1+\left[\frac{-16ga_w v_w^3}{u_{*w}^4 k_w^2 \rho_w c_w}(Q+f_s\,SW)\right]^{3/4}\right\}^{-1/3} \tag{3}$$

$$\delta = 6\left\{1+\left[\frac{-16ga_w v_w^3}{u_{*w}^4 h_w^2 \rho_w c_w}(Q+f_s\,SW)\right]^{3/4}\right\}^{-1/3} \tag{3a}$$

In the new definintions, the relationship between heat conductivity and heat diffusivity is $k_w = h_w \cdot \rho_w \cdot c_w$. If we multiply both the numerator and the denominator by '$\rho_w \cdot c_w$', respectively:

$$\delta = 6\left\{1+\left[\frac{-16g\rho_w c_w a_w v_w^3}{u_{*w}^4 h_w^2 \rho_w^2 c_w^2}(Q+f_s\,SW)\right]^{3/4}\right\}^{-1/3} \tag{3b}$$

Because $k_w = h_w \cdot \rho_w \cdot c_w$, Eq. (3b) becomes Eq. (4):

$$\delta = 6\left\{1+\left[\frac{-16g\rho_w c_w a_w v_w^3}{u_{*w}^4 k_w^2}(Q+f_s\,SW)\right]^{3/4}\right\}^{-1/3} \tag{4}$$

Therefore, our transition from Eq. (3) to Eq. (4) is correct.

*7. The change in the modified model produces a (desired) smoother simulated $T_{int}$ compared*

*with the original ECMWF (fig 5-6), however, the magnitude also seems to be sufficiently changed. It would be good to add some discussion on why this might be and what are the implications. Also, please clarify whether $T_{fnd}$ is expected to change based on the new model.*

The model employed by ECMWF is based on incorrect equations that lack unit consistency, resulting in a fundamentally flawed simulation process. Despite the unit inconsistency, the model equations may have been coded for computer calculations with incorrect coefficients assigned to produce the results. As discussed in sections #5–6, this approach can effectively nullify the function of the cool skin layer. This interpretation is supported by the observation that ERA5 $T_s$ does not respond well to solar radiation flux (Fig. 9). In accurate simulations, the diurnal variation of $T_{cool}$ should be smaller than $T_s$, meaning the diurnal variation of the current ERA5 $T_s$ should be smaller than the new ones. Moreover, the ERA5 $T_s$ diurnal variation may not follow the solar flux variation, which directly influences the $T_s$.

Due to the ineffective cool skin layer in ERA5, it is likely that the skin temperature variation follows the variation of the warm layer, which is more complex due to the competing influences of solar radiation and turbulent mixing. However, given that the model development was based on incorrect equations that lack unit consistency, further discussion on how the temperature within warm layer is obtained may not be worthwhile.

In the revised version, following discussion is added in the conclusion and discussion section.

"The unreasonable features noted in ERA5 $T_s$ may be attributed to the failure to simulate the temperature within the cool skin layer due to the incorrect equation and the wrong assignment of diffusivity. In addition to the unit inconsistency, the wrong diffusivity value (i.e., 0.6 instead of $1.40 \times 10^{-7}$ $m^2$ $s^{-1}$) may result in $T_s$ being nearly equal to $T_{cool}$ (i.e., $T_s \approx T_{cool}$). However, in reality, as expressed in Eq. (2), $T_s$ variation should be much larger than $T_{cool}$ during the daytime due to incoming solar flux. Such expected difference is confirmed by observing that ERA5 $T_s$ is not well responsive to solar radiation flux (Fig. 9). Since the diurnal variation of $T_{cool}$ should be smaller than $T_s$ in accurate simulations, the diurnal variation of the current ERA5 $T_s$ should be smaller than the new ones, as revealed in Fig. 9. Furthermore, the ERA5 $T_s$ diurnal variation may not follow the solar flux variation, whose direct influence is on $T_s$. Because the cool layer physics was effectively suppressed in ECMWF model (as shown in $T_s \approx T_{cool}$), the ERA5 $T_s$ variation may follow the variation of the warm layer, which is more complex due to the competing influences of solar radiation and turbulent mixing. This might result in more irregular patterns as shown in Fig. 9. It appears that the problems in the current ECMWF may not be confined to the assignment of wrong values for the heat diffusivity, but also may reside in the warm layer temperature variation. It is because the warm layer process might have been developed based on incorrect equations of the cool skin process."

Dependence of $T_{fnd}$ on the model

$T_{fnd}$ (now referred to as $T_{warm}$ in the revised version) is not model-dependent, but rather an observation-based SST temperature. The model uses this observation-based SST as the

temperature at the bottom of the warm layer ($T_{warm}$). Different models may assign $T_{fnd}$ as $T_{warm}$ at different levels. This discussion is also relevant for answering #3.

In the revised version, the following sentences have been found:

"The temperature at the bottom of the warm layer ($T_{warm}$) is assumed to have no diurnal variation, representing the model's bottom boundary condition. It is updated daily with observation-based temperature and remains fixed until the next update. Note that $T_{warm}$ is equivalent to ERA5 SST."

*8. It is hard to assess whether a mean deviation of 0.1 K and standard deviation of 0.5 K is a strong result given how sensitive $T_{int}$ is (e.g., line 45 in the introduction). Can you compare with a different baseline model here to make this claim stronger?*

To convey the significance of the error statistics in this study, it is more effective to compare them with observational errors rather than with other model results. This is because observational data can serve as a reliable reference. Fortunately, the model results were compared with M-AERI and 1-m depth buoy data, and there are existing comparisons between the satellite-derived SST and the same M-AERI and buoy observations. By comparing the current error statistics with the errors from the satellite observations, we can better shed light on the significance of the current error statistics.

Error statistics for three satellite SST products based on longwave nonlinear algorithms (Aqua-MODIS, Terra-MODIS, and S-NPP-VIIRS) compared to M-AERI and buoy observations are presented in Tables 1 and 2, respectively. It is important to note that SSTs from these three satellites are derived from IR measurements, which represent the temperature at a depth similar to the IR-based M-AERI (~10-μm depth), but differ from the 1-meter depth buoy-observed SSTs. Consequently, a larger bias is expected when comparing IR-based satellite products with buoy observations. As expected, IR-based satellite products show a larger bias against buoy observations, which is normally corrected when buoy-like SSTs are produced from satellite measurements. With this in mind, we conclude that the errors of model-produced temperatures at two levels are similar to those of satellite observations. Therefore, the model-produced skin temperature in this study would be comparable to satellite measurements, if such satellite data were available.

**Table 1.** Statistics of errors in SSTs at the sub-skin layer over global ocean: comparison of this study and satellite sensor-retrieved temperatures vs. M-AERI-measured temperatures for best quality level only (NASA ATBD; https://oceancolor.gsfc.nasa.gov/resources/atbd/sst/#sec_4, last accessed on July 24, 2024).

| SST | Mean Deviation | Standard Deviation | Count |
| --- | --- | --- | --- |
| **This study** | **-0.08 K** | **0.49 K** | **32,647** |
| Terra MODIS | −0.06 K | 0.48 K | 3,069 |

| SST | Mean Deviation | Standard Deviation | Count |
|---|---|---|---|
| Aqua MODIS | 0.04 K | 0.49 K | 2,070 |
| S-NPP VIIRS | 0.03 K | 0.20 K | 81 |

**Table 2.** Same as in Table 1 except for 1-m depth buoy-measured temperature.

| SST | Mean Deviation | Standard Deviation | Count |
|---|---|---|---|
| **This study** | **-0.07 K** | **0.28 K** | **6,241,008** |
| Terra MODIS | −0.17 K | 0.44 K | 538,918 |
| Aqua MODIS | −0.19 K | 0.42 K | 508,950 |
| S-NPP VIIRS | −0.21 K | 0.48 K | 473,498 |

Accordingly, in the revised version, we have newly added the following phrases, together with Tables 1 and 2:

"To effectively convey the significance of the error statistics in this study, we compared them with error statistics in satellite products. The model results were compared with M-AERI and 1-m depth buoy data, and existing comparisons between satellite-derived SSTs and the same observations were utilized. Error statistics for three satellite SST products based on the longwave nonlinear algorithm (Aqua-MODIS, Terra-MODIS, and S-NPP-VIIRS) against M-AERI and buoy observations are presented in Tables 1–2, alongside the model results. It is important to note that SSTs from these three satellites are derived from IR measurements, which represent the temperature at a depth similar to the IR-based M-AERI (~10-μm depth) but differ from the 1-m depth buoy-observed temperature. Consequently, IR-based satellite products show a large bias against buoy observations, which is normally corrected when buoy-like SSTs are produced from satellite measurements. From these comparisons, we conclude that the errors of model-produced temperatures at two levels are similar to those of satellite products."

*9. The conclusion would benefit from additional discussion on the implications of this work, a comparison with other models, and what might the next steps be.*

Thank you for your feedback. In the revised version, we have provided an expanded discussion on the benefits of this work in the conclusion and discussion section and what might be next step, as follows:

"The enhanced forecasting capability of the diurnal cycle of $T_s$, whose accuracy is comparable to satellite observations, should be beneficial across a wide range of meteorological applications and further contribute to related climate and weather studies. Since the diurnal cycle of $T_s$ affects various factors such as surface net heat flux, surface wind, evaporation rate, and atmospheric stability, this simulation of $T_s$ would improve weather forecasting accuracy. However, these expectations need to be confirmed through NWP model experiments using the revised OML model to examine how the improved $T_s$ influences predicted variables. Therefore,

future studies should focus on conducting NWP experiments to understand the potential benefits and limitations of the revised OML model in meteorological applications."

*10. It was not clear to me how the ATLAS data was used to derive the stability functions based on the current explanation in the appendix. It may be helpful to add an additional section on this topic to accompany the figures shown in the main manuscript.*

In the revised version, we added more explanation on how the stability functions are obtained. You asked this topic presented in the main text as a separate section. However, considering that numerous terms and coefficients are defined and given before explaining how stability functions are developed, we decided to separate this topic from Appendix A and put in new Appendix B. In this way, readers can understand procedures continuously from Appendix A to Appendix B.

In the revised version, Appendix B is presented as follows:

"The stability functions ($\phi$; Eqs. A10, A11, and A15) are formulated using data from ATLAS buoys deployed at 0° and 140°W from 2001 to 2020. These buoys measure temperatures at a depth of 1 meter. Based on the relation obtained by directly comparing OSTIA data (i.e., ERA5 SST) with buoy data, their mean temperature is treated as $T_{warm}$ with non-diurnal cycle. ERA5 atmospheric forcing data are also collected. By applying the predefined temperature profile shape function (Eq. A1), $T_{cool}$ can be derived from the $T_{1m}$ and $T_{warm}$ values. This process enables the $T_{cool}$ and $T_{warm}$ dataset to be organized into a specific time interval ($\Delta t = 3600$ seconds) as n, n+1, n+2, and so forth.

The magnitude of $\phi$ is calculated under the two conditions proposed by Zeng and Beljaars (2005). The first condition (Case 1) is when relatively higher temperatures persist for a certain period after sunset (i.e., $T_{cool} - T_{warm} \geq 0$ and $Q + f_d\ SW < 0$). The second condition (Case 2) covers all other scenarios.

$$\text{Case 1:} \quad \phi^{(n)} = \left| -\frac{\beta u_{*w}^{(n)} \Delta t \left( T_{cool}^{(n)} - T_{warm}^{(n)} \right)}{d \left( T_{cool}^{(n+1)} - T_{warm}^{(n)} \right)} \right| \tag{B1}$$

$$\text{Case 2:} \quad \phi^{(n)} = \left| -\frac{\beta u_{*w}^{(n)} \Delta t \left( T_{cool}^{(n)} - T_{warm}^{(n)} \right)}{d \left( T_{cool}^{(n+1)} - T_{warm}^{(n)} \right) - \Delta t \alpha \left( Q^{(n)} + f_d\ SW^{(n)} \right)} \right| \tag{B2}$$

The corresponding stability parameters ($\zeta$) are calculated using Eq. (A12). The scatterplots of buoy-derived $\zeta$ vs. $\phi$ are presented in Fig. B1, demonstrating relationships similar to those proposed by Stiperski and Calaf (2023). As a result, the fitted $\phi$–$\zeta$ relationship are determined for three distinct cases—Case 1 (for Eq. A15) and Case 2, which is further divided into $\zeta > 0$ (for Eq. A10) and $\zeta \leq 0$ (for Eq. A11)."

**Minor comments**

*M1. Line 12: replace "and" after "ECMWF with comma.*

In accordance with #1, during the comprehensive rewriting of the manuscript, this sentence has been removed.

*M2. Line 45: replace "evidenced" with "evident".*

In the revised version, we have replaced the word "evident" with "evidenced"

*M3. Lines 47-48: this last sentence is out of context. You haven't told the reader that models have problems with heat budgets. Also, what are related phenomena? Please rephrase.*

It is now stated as follows: "Capturing this diurnal variability is essential, as skin temperature governs moisture, heat, and radiation fluxes that dictate ocean-atmosphere interactions. Consequently, accurate modeling of diurnal variation in skin temperature is important for studying heat budgets and related phenomena."

*M4. Line 54: odd sentence, please reword.*

Old: "However, as the representing depths of ERA5 skin temperature and other observations are different, the direct comparison and deduced results cannot be overly credited."

New: "However, since ERA5 skin temperature and in-situ observations represent different depths, results derived from direct comparisons should be interpreted with caution."

*M5. Line 57: what deficiencies? Can you be more clear? Are these based on figures 1-2? If so the reader hasn't seen them yet.*

In accordance with #1, during the comprehensive rewriting of the manuscript, this sentence has been removed.

*M6. Lines 58-61: please reword this entire paragraph as it is currently without context.*

Old: "Recognizing potential problems in the used scheme for generating the ocean skin temperatures, we attempt to scrutinize the root cause of anomalous features found in ERA5 (as exhibited in the following section 2), and revise the Zeng and Beljaars (2005) scheme for the better simulations of the diurnal variation of ocean skin temperature. Results and lessons obtained in this study will eventually lead to better weather forecasting and climate simulation."

New: "Recognizing potential issues in generating ocean skin temperatures in the ECMWF model, we scrutinize the errors in the Zeng and Beljaars (2005) scheme and attempt to revise it. We then examine the impact of the revised scheme on the diurnal variation of ocean skin temperature and assess the accuracy of the corrected skin temperature. The results and insights gained from this study aim to improve weather forecasting and climate simulation."

*M7. First paragraph in section 2a may be better suited for the intro.*

Considering that the various definitions and abbreviations of SST may have caused confusion among readers (see #2), we have removed subsection 2a, spanning Lines 64 to 84, and simplified the overall terminology. Accordingly, the relevant temperatures are defined at the point of use.

In line with M7, subsection 2a, spanning Lines 64 to 84, has been removed in the revised version.

Those acronyms are removed in the revised version.

*M10. Line 78: there must be some signal of the diurnal cycle below 5-10 m as the mixed layer is sensitive to changes in heat flux and wind stress. Please clarify.*

The relevant response has been provided in #3. Furthermore, subsection 2a, spanning Lines 64 to 84, has been removed in the revised version.

In line with M7, subsection 2a, spanning Lines 64 to 84, has been removed in the revised version. Additionally, in the main text, the term 'conventional measurements' has been replaced with 'buoy/shipborne measurements.'

For example: "Current methods for constructing SST data across the global oceans, available on a daily time scale, incorporate satellite-borne radiometric measurements and buoy/shipborne measurements (Reynolds and Smith, 1994; Ishii et al., 2005; Reynolds et al., 2007; Donlon et al., 2012; Titchner and Rayner, 2014)."

*M12. Line 80: Can you expand on the type of bulk models used in NWP? It is important in the context of the vertical profile you are alluding to in finding a better SST$_{fnd}$.*

Types of models employed for simulating $T_s$ are typically classified into diffusion models, bulk (or slab) models, and empirical parametric models. Additionally, bulk models can be further subdivided into multilayer and simplified models. The OML model used in this study is a simplified two-layer slab model. In some numerical models, the observational SST is fixed as $T_s$ without using these models (not considering the diurnal variation).

*M13. Line 80: replace "deal" with "examine" and remove the following "with".*

In line with M7, subsection 2a, spanning Lines 64 to 84, has been removed in the revised version.

*M14. Lines 90-91: I believe you are comparing different times in UTC and not time zones (which will refer to longitudinal differences)?*

All used data are provided in UTC.

*M15. Line 94: please reword the description of figure 1.*

Old: "Specifically, the hourly variation in Fig. 1e–1h exhibits spatially much variating patterns, contrasting to much smoother features at earlier and later 12-hour periods across the global ocean."

New: "Specifically, the hourly variations in Fig. 4e–h exhibit patterns with spatially incoherent, localized variations, in contrast to the smoother features observed during the earlier and later 12-h periods (Fig. 4a–d and 4i–l) across the global ocean."

*M16. Line 105: can winds no affect SST on the order of one hour? Is this not represented in the ECMWF model?*

In the OML model, winds influence SST, as indicated by the last term on the right-hand side of Eq. (A8). However, the original model may not fully capture this influence due to potential errors – see #7.

*M17. Can you show an equivalent of figure 7 for the original ECMWF model for comparison?*

The ERA5-original $T_s$ is already depicted by a blue solid line in Fig. 9 (previously Fig. 7).

*M18. Lines 139: please provide references for the replacement of k_w with thermal diffusivity.*

The relevant response has been provided in #5–6.

*M19. Line 155: replace "it turned out" with something less casual.*

In the revised version, we have replaced "it turned out that" with "it was found that."

*M20. Lines 199-205: very confusing paragraph. Please reword and clarify procedure.*

We have rewritten it as follows:

"The M-AERI data, spanning an 8-year period from 2013 to 2020, are collected from 24 voyages (Minnett et al., 2020). Of these, 2 voyages from 2020 are used to determine the scaling factor for parameter λ (detailed in Appendix A), while the remaining 22 voyages from 2013 to 2019 are used for model validation. Details of the individual cruises, vessels, and their tracks can be found in Fig. S1 and Table S1. We select only measurements with an uncertainty range within 0.1 K (Minnett et al., 2005) and at least 25 km away from the coastline. The data are then thinned to 1-h intervals to align with the model's temporal resolution. Additionally, a spatial collocation procedure is performed, mapping data points to the nearest model grid points."

In the revised version, we have replaced the word "parameterizations" with "stability functions"

*M22. Line 230: please be more specific with the term "statistics".*

Old "The comparison results are exhibited in Fig. 3 with statistics. The comparison shows a good agreement with a correlation coefficient of 0.99, a mean deviation of −0.08 K, and a standard deviation of 0.49 K for a total of 32,647 data points."

New: "The comparison, illustrated in Fig. 2, shows statistical results of a correlation coefficient of 0.99, a mean deviation of −0.08 K, and a standard deviation of 0.49 K from a total of 32,647 data points."

*M23. Line 251-252: please reword*

In the revised version, the following change has been made:

Old: "Thus, it is presumed that diurnal variations of ERA5 $T_{int}$ are problematic, as two different alternating regimes every 12 hours should not be reasonable. We cannot attribute any atmospheric or oceanic forcings that would give rise to the periodic oscillations in the $T_{int}$ distribution with a 12-hour cycle."

New: "Thus, it is presumed that the diurnal variations of ERA5 $T_s$ are problematic, as the presence of alternating regimes every 12 hours is not reasonable. There are no identifiable forcings that would cause periodic oscillations in the $T_s$ variation distribution over the global ocean with a 12-h cycle."

*M24. Line 266: Do you mean ERA5 here or the new prediction of Tint?*

We think that in the original manuscript, the sentence in Line 266 lacked coherence with the preceding and following sentences. Therefore, it has been removed in the revised version.

---

## Author Comment (AC3)

**Response to Referee 2**

*The manuscript titled "A revised ocean mixed layer model for better simulating the diurnal variation of ocean skin temperature" focuses on the fine-tuning and validation of an ocean mixed layer model (OMLM) used at ECMWF. The authors initially describe the shortcomings of the existing OMLM, then rectify the typographical errors in the model, and subsequently validate the results.*

**Comments and suggestions**

*1. However, the manuscript, in its current form, lacks proper organization and sequence. The methodology is described before the dataset is introduced, leading to confusion. Various SST definitions and terminologies are introduced without context, puzzling the reader. The authors should first provide a detailed description of the data used, including sources, resolution, etc., followed by the methodology. This should include defining various SST terms and explaining each term's source from the dataset.*

Manuscript organization and sequence:

Thanks for the suggestion. We initiated this work upon discovering that the ERA5 skin temperature data exhibited erroneous features, particularly the changing spatial patterns at 10 UTC and 22 UTC. Further investigation revealed errors in the prognostic scheme employed by the ECMWF model. As a result, our initial presentation in the previous version focused on these erroneous features before detailing our corrective measures, which may have seemed awkward. Other reviewer also suggested restructuring the manuscript.

Following your suggestion, in the revised version, we introduced the data set used after the introduction, identified model errors, detailed the subsequent revisions, validated the revised model, and presented the results, including the erroneous features in the ERA5 data.

Recognizing that the various definitions and abbreviations of SST may have caused confusion, we defined only the relevant temperatures at their points of use. We removed the SST definition section (previously subsection 2a) and simplified the overall SST terminology in the revised manuscript. A new Fig. 1 visually presents the SST terms and aids in understanding the OML model structure. Thus, all temperature-related definitions/acronyms are limited to five shown in Fig. 1: $T_s$, $T_{cool}$, $T_{warm}$, $T_z$, and SST. Other IR and buoy-measured temperatures are referred to as the temperatures at 10 μm and 1 m depths, respectively, without additional acronyms.

[Figure]

**Figure 1.** Schematic diagram illustrating the ocean mixed layer (OML) model. $T_s$, $T_{cool}$, and $T_{warm}$ represent the temperatures at three levels of the OML model. In ERA5, skin temperature and SST correspond to $T_s$ and $T_{warm}$, respectively. Temperature $T_z$ represents the temperature at depth z. The thick solid lines depict the temperature profiles as expressed by two equations in the cool skin and warm layers.

*2. Figures 3 and 4 should also include comparisons with the original ECMWF model. Although the authors state that the revised model has the same error range, it would be beneficial to compare the scatter plots from the original model for clarity.*

It would be ideal to make a direct comparison between $T_s$ from the original ERA5 and observations, as shown in Figs. 3 and 4. Note that the two temperatures at two depths (10 μm and 1 m) used for comparison against IR and buoy observations in Figs. 3 and 4 are from the OML model-based temperature profile within the cool skin and warm layers. However, since the ECMWF model does not save the intermediate output (here ocean temperature profile), a direct comparison is not possible.

Although direct comparison is not allowed, significant underestimates of $T_s$ by ECMWF are evident in Fig. 9. We have now explained this in the revised version. The ECMWF model incorrectly used thermal diffusivity as 0.6 instead of approximately $1.40 \times 10^{-7}$, in addition to a unit inconsistency. Thus, the denominator on the right-hand side of Eq. (2) is roughly $4 \times 10^6$ times larger than the correct value. Additionally, the incorrect assignment of thermal diffusivity affects the depth of the cool skin layer (δ). Our calculations indicate that the wrong coefficient

causes $\delta$ to be approximately $1\times10^{-4}$–$3\times10^{-4}$ times that of the correct model. Overall, the incorrect assignment induces "$T_s - T_{cool}$" to be about $10^{-3}$ of the expected values, approaching near zero (i.e. $T_s \approx T_{cool}$) regardless of the heating magnitudes at the surface.

$$T_s - T_{cool} = \frac{\delta}{h_w \rho_w c_w}\left(Q + f_s\, SW\right) \tag{2}$$

This effectively eliminates the role of cool skin layer in the diurnal variation, resulting in $T_s$ being equal to $T_{cool}$, leading to erroneous $T_s$ simulations in the ECMWF model. This explanation is now included in the conclusion and discussion section as follows:

"The unreasonable features noted in ERA5 $T_s$ may be attributed to the failure to simulate the temperature within the cool skin layer due to the incorrect equation and the wrong assignment of diffusivity. In addition to the unit inconsistency, the wrong diffusivity value (i.e., 0.6 instead of $1.40\times10^{-7}$ $m^2$ $s^{-1}$) may result in $T_s$ being nearly equal to $T_{cool}$ (i.e., $T_s \approx T_{cool}$). However, in reality, as expressed in Eq. (2), $T_s$ variation should be much larger than $T_{cool}$ during the daytime due to incoming solar flux. Such expected difference is confirmed by observing that ERA5 $T_s$ is not well responsive to solar radiation flux (Fig. 9). Since the diurnal variation of $T_{cool}$ should be smaller than $T_s$ in accurate simulations, the diurnal variation of the current ERA5 $T_s$ should be smaller than the new ones, as revealed in Fig. 9. Furthermore, the ERA5 $T_s$ diurnal variation may not follow the solar flux variation, whose direct influence is on $T_s$. Because the cool layer physics was effectively suppressed in ECMWF model (as shown in $T_s \approx T_{cool}$), the ERA5 $T_s$ variation may follow the variation of the warm layer, which is more complex due to the competing influences of solar radiation and turbulent mixing. This might result in more irregular patterns as shown in Fig. 9. It appears that the problems in the current ECMWF may not be confined to the assignment of wrong values for the heat diffusivity, but also may reside in the warm layer temperature variation. It is because the warm layer process might have been developed based on incorrect equations of the cool skin process."

*3. The manuscript does not present any significant scientific advancements in the present form beyond correcting typographical errors in the original model. While the authors have contributed by identifying and fixing these mistakes, this could have been addressed in a technical internal note rather than a peer-reviewed publication. Furthermore, the practical implications of the errors in the mixed layer model are not clearly articulated, aside from the improper simulation of diurnal variability. The authors should emphasize the practical benefits gained from correcting the OMLM.*

We respectfully disagree with your opinion that this paper merely addresses typographical errors in the original model. As detailed in the revised model section, especially in the Appendix, the corrections are far more extensive than simple typographical fixes. Correcting the erroneous equations used in the original model is only part of the revision process. Since the original model was developed based on incorrect equations, which do not even have unit consistency, the entire simulation process was flawed. As discussed in #2, the current ECMWF version seems to have no functioning cool skin layer. Therefore, rectifying the identified errors required a comprehensive overhaul of the model, including the development of corrected equations, model configuration, and new stability functions.

Additionally, we have included a new discussion on why ERA5 shows significant underestimates compared to $T_s$ simulations with the revised model, as discussed in #2. Further, we provided error statistics for $T_s$, demonstrating the reliability of these values. By comparing the temperatures obtained from simulations against IR-based SST retrievals and buoy observations, we were able to demonstrate that $T_s$ error statistics are comparable to errors in satellite measurements of SST. This study presents the first instance of such error statistics for $T_s$ diurnal variation obtained from model simulations.

To convey the importance of these error statistics, we compared them with observational errors, as observational data serve as a reliable reference. In this study, the model results were compared with M-AERI and 1-m depth buoy data. Existing comparisons between satellite-derived SST and the same M-AERI and buoy observations allow us to understand the significance of the current error statistics better. Error statistics for three satellite SST products based on longwave nonlinear algorithm (Aqua-MODIS, Terra-MODIS, and S-NPP-VIIRS) compared to M-AERI and buoy observations are presented in Tables 1 and 2, respectively. It is important to note that SSTs from these three satellites are derived from IR measurements, which represent the temperature at a depth similar to the IR-based M-AERI (~10-μm depth), but differ from the 1-m depth buoy-observed SSTs. Consequently, a larger bias is expected when comparing IR-based satellite products with buoy observations. As expected, IR-based satellite products show a larger bias against buoy observations, which is typically corrected when buoy-like SSTs are produced from satellite measurements.

Considering this, we conclude that the errors of model-produced temperatures at two levels are similar to those of satellite observations. Therefore, the model-produced skin temperature in this study would be comparable to satellite measurements, if such satellite data were available.

In conclusion, this work is not a mere correction of typographical errors but involves developing a corrected model whose $T_s$ simulation accuracies are comparable to satellite observations.

We added following sentence in the subsection 3a:

"With the corrections and modifications mentioned above, the implemented OML model is referred to as the "revised" OML model in this study. Correcting the erroneous equations used in the original model, which do not show even unit consistency, is only part of the revision process. Since the original model was developed based on incorrect equations, the entire simulation process was also considered flawed. Therefore, rectifying the identified errors requires a comprehensive rebuilt of the model, including the development of corrected equations, model configuration, and new calculation of stability functions with revised model. A detailed description of the model structure and configuration can be found in Appendices A–B."

**Table 1.** Statistics of errors in SSTs at the sub-skin layer over global ocean: comparison of this study and satellite sensor-retrieved temperatures vs. M-AERI-measured temperatures for best quality level only (NASA ATBD; https://oceancolor.gsfc.nasa.gov/resources/atbd/sst/#sec_4, last accessed on July 24, 2024).

| SST | Mean Deviation | Standard Deviation | Count |
| --- | --- | --- | --- |

| | | | |
|---|---|---|---|
| **This study** | **-0.08 K** | **0.49 K** | **32,647** |
| Terra MODIS | −0.06 K | 0.48 K | 3,069 |
| Aqua MODIS | 0.04 K | 0.49 K | 2,070 |
| S-NPP VIIRS | 0.03 K | 0.20 K | 81 |

**Table 2.** Same as in Table 1 except for 1-m depth buoy-measured temperature.

| SST | Mean Deviation | Standard Deviation | Count |
|---|---|---|---|
| **This study** | **-0.07 K** | **0.28 K** | **6,241,008** |
| Terra MODIS | −0.17 K | 0.44 K | 538,918 |
| Aqua MODIS | −0.19 K | 0.42 K | 508,950 |
| S-NPP VIIRS | −0.21 K | 0.48 K | 473,498 |